# Hospital admission with non-alcoholic fatty liver disease is associated with increased all-cause mortality independent of cardiovascular risk factors

**Jake P. Mann**[1,2,3]*, **Paul Carter**[3,4], **Matthew J. Armstrong**[5], **Hesham K. Abdelaziz**[6,7], **Hardeep Uppal**[3], **Billal Patel**[6], **Suresh Chandran**[8], **Ranjit More**[6], **Philip N. Newsome**[9,10]☉, **Rahul Potluri**[3]☉

**1** MRC Epidemiology Unit, Institute of Metabolic Science, University of Cambridge, Cambridge, United Kingdom, **2** Department of Paediatrics, University of Cambridge, Cambridge, United Kingdom, **3** ACALM Study Unit in collaboration with Aston Medical School, Aston University, Birmingham, United Kingdom, **4** Division of Cardiovascular Medicine, Department of Medicine, University of Cambridge, Cambridge, United Kingdom, **5** Liver Unit, University Hospitals Birmingham NHS Foundation Trust, Birmingham, United Kingdom, **6** Lancashire Cardiac Centre, Blackpool Victoria Hospital, Blackpool, United Kingdom, **7** Department of Cardiovascular Medicine, Ain Shams University Hospital, Cairo, Egypt, **8** Department of Medicine, Pennine Acute Hospitals NHS Trust, Manchester, United Kingdom, **9** National Institute for Health Research Liver Biomedical Research Unit at University Hospitals Birmingham NHS Foundation Trust and the University of Birmingham, Birmingham, United Kingdom, **10** Centre for Liver Research, Institute of Immunology and Immunotherapy, University of Birmingham, Birmingham, United Kingdom

☉ These authors contributed equally to this work.
* jm2032@cam.ac.uk

**Data Availability Statement:** All relevant data are within the manuscript and its Supporting Information files.

## Abstract

Non-alcoholic fatty liver disease (NAFLD) is common and strongly associated with the metabolic syndrome. Though NAFLD may progress to end-stage liver disease, the top cause of mortality in NAFLD is cardiovascular disease (CVD). Most of the data on liver-related mortality in NAFLD derives from specialist liver centres. It is not clear if the higher reported mortality rates in individuals with non-cirrhotic NAFLD are entirely accounted for by complications of atherosclerosis and diabetes. Therefore, we aimed to describe the CVD burden and mortality in NAFLD when adjusting for metabolic risk factors using a 'real world' cohort. We performed a retrospective study of patients followed-up after an admission to non-specialist hospitals with a NAFLD-spectrum diagnosis. Non-cirrhotic NAFLD and NAFLD-cirrhosis patients were defined by ICD-10 codes. Cases were age-/sex-matched with non-NAFLD hospitalised patients. All-cause mortality over 14-years follow-up after discharge was compared between groups using Cox proportional hazard models adjusted for demographics, CVD, and metabolic syndrome components. We identified 1,802 patients with NAFLD-diagnoses: 1,091 with non-cirrhotic NAFLD and 711 with NAFLD-cirrhosis, matched to 24,737 controls. There was an increasing burden of CVD with progression of NAFLD: for congestive heart failure 3.5% control, 4.2% non-cirrhotic NAFLD, 6.6% NAFLD-cirrhosis; and for atrial fibrillation 4.7% control, 5.9% non-cirrhotic NAFLD, 12.1% NAFLD-cirrhosis. Over 14-years follow-up, crude mortality rates were 14.7% control, 13.7% non-cirrhotic NAFLD, and 40.5% NAFLD-cirrhosis. However, after adjusting for demographics,

**Funding:** JPM is supported by a Wellcome Trust Fellowship (216329/Z/19/Z, https://wellcome.ac.uk). The funders had no role in study design, data collection and analysis, decision to publish, or preparation of the manuscript.

**Competing interests:** The authors have declared that no competing interests exist.

**Abbreviations:** ACALM, Algorithm for Comorbidities Associations Length of stay and Mortality; CHF, congestive heart failure; CVD, cardiovascular disease; HCC, hepatocellular carcinoma; HR, hazard ratio; ICD-10, International Classification of Disease, 10th edition; NAFL, non-alcoholic fatty liver; NAFLD, non-alcoholic fatty liver (disease); NASH, non-alcoholic steatohepatitis; NIDDM, non-insulin dependent diabetes mellitus; NHS, National Health Service (UK); OPCS-4, Office of Population Censuses and Surveys Classification of Interventions and Procedures.

non-cirrhotic NAFLD (HR 1.3 (95% CI 1.1–1.5)) as well as NAFLD-cirrhosis (HR 3.7 (95% CI 3.0–4.5)) patients had higher mortality compared to controls. These differences remained after adjusting for CVD and metabolic syndrome components: non-cirrhotic NAFLD (HR 1.2 (95% CI 1.0–1.4)) and NAFLD-cirrhosis (HR 3.4 (95% CI 2.8–4.2)). In conclusion, from a large non-specialist registry of hospitalised patients, those with non-cirrhotic NAFLD had increased overall mortality compared to controls even after adjusting for CVD.

## Introduction

Non-alcoholic fatty liver disease (NAFLD) is the most common liver disease in Europe [1] and is strongly associated with all features of the metabolic syndrome [2]. The majority of patients with NAFLD have simple steatosis (non-alcoholic fatty liver, NAFL) and only a minority have non-alcoholic steatohepatitis (NASH), with or without fibrosis. However a small, but significant proportion do progress to end-stage liver disease [3].

NAFLD is thought to be associated with increased all-cause and cardiovascular mortality [4–6]. It has been established that fibrosis is the main predictor of long-term liver-related morbidity in NAFLD [3, 7–9] and that patients with NASH but no fibrosis have a similar outcome to those with NAFL and no fibrosis. However, these studies included biopsy-proven patients in specialist clinics and therefore there is likely significant ascertainment bias in estimating rates of hepatic complications. Recent data also suggests that fibrosis stage is a critical predictor of cardiovascular events in NAFLD [10]. The natural history of NAFLD and its impact upon clinical services is an important topic that divides expert opinion [11, 12].

Cardiovascular disease (CVD) is the commonest cause of mortality in patients with NAFLD [3]. A recent large-scale analysis strongly suggests that this is due to prevalence of classical CVD risk factors such as type 2 diabetes and dyslipidaemia [13]. Insulin resistance is understood to be the primary driver linking all these features of the metabolic syndrome. In response to the positive energy balance of obesity, subcutaneous adipose becomes dysfunctional and there is expansion of visceral white adipocytes, which are less insulin sensitive and have a higher basal rate of lipolysis [14]. Elevated insulin and increased substrate delivery to the liver promotes hepatic steatosis by driving increased *de novo* lipogenesis without increasing glucose uptake [15]. Cumulatively, this results in a rise in circulating triglycerides, impaired low-density lipoprotein clearance, and higher serum glucose. Hepatic steatosis is also thought to alter the composition of secreted lipoparticles [16].

A further increasingly important consideration is the burden of cardiovascular co-morbidity in patients with end-stage NAFLD for whom transplantation is an option [17]. There is currently comparatively limited data on the prevalence of CVD in patients with NAFLD cirrhosis [18]. One study of 133 patients with cryptogenic cirrhosis found nearly half to have diabetes and 18% had experienced a major cardiovascular or cerebrovascular event [19]. CVD events are common post-transplant sequelae and chronic kidney disease is linked to reduced graft survival [20].

Whilst several previous natural history studies have included comparison to age- and gender-matched control populations, they have been unable to control for CVD [21–23]. Therefore, it remains unclear whether NAFLD is associated with increased all-cause mortality after correction for cardiac and metabolic disease risk factors.

We aimed to first describe the burden of CVD across the NAFLD disease spectrum: non-cirrhotic-NAFLD and NAFLD-cirrhosis (end-stage fibrosis). Then to assess whether they are associated with increased all-cause mortality in a real life cohort of hospitalised UK patients

from the ACALM (Algorithm for Comorbidities, Associations, Length of stay and Mortality) registry, after correction for CVD and metabolic risk factors.

## Materials and methods

### Study design

The study was conducted as a retrospective cohort study of adult patients in England during 2000–2013 who were admitted to 7 different hospitals with naturalistic follow-up. All available data was included. Tracing of anonymised patients was performed using the ACALM (Algorithm for Comorbidities, Associations, Length of stay and Mortality) study protocol to develop the ACALM registry and has been previously described by our group [24–29]. Briefly, medical records were obtained from local health authority computerized Hospital Activity Analysis register, which is routinely collected by all NHS hospitals. This provides fully anonymized data on hospital admissions and allows for the long-term tracing of patients at an individual hospital. The ACALM protocol uses the International Classification of Disease, 10<sup>th</sup> edition (ICD-10) and Office of Population Censuses and Surveys Classification of Interventions and Procedures (OPCS-4) coding systems to trace patients. This data was obtained separately for the seven included hospitals. Similar data could be obtained through national Hospital Episode Statistics or from any local Hospital Activity Analysis register.

### Participant identification

ICD-10 codes were used to identify patients with NAFL (non-alcoholic fatty liver, K76.0), NASH (non-alcoholic steatohepatitis, K75.8), and NAFLD-cirrhosis (cryptogenic cirrhosis, K74.6). Where a patient was coded with both NAFL and NASH, they were included in the NASH group. Patients coded with both NAFL and NAFLD-cirrhosis, or NASH and NAFLD-cirrhosis, were included in the cirrhosis group. NAFL and NASH groups were combined to create a non-cirrhotic-NAFLD group. As per UK practice, the diagnosis of NAFL, NASH, or NAFLD-cirrhosis were made according to clinical judgement and the latest guidelines but the results of the investigations used to derive the diagnoses were not available. An age- and sex-matched control group (with no liver-related diagnoses) was identified from the same ACALM registry and matched 10:1 to patients with NAFLD diagnoses. In line with recent recommendations [30], patients with a history of alcohol excess (F10) or any other alcohol-related diagnosis were excluded. In addition, patients with other liver diseases were excluded, including: autoimmune hepatitis (K75.4), viral hepatitis (B15-B19), Wilson disease and haemochromatosis (E83), cholangitis (K83.0) and primary biliary cirrhosis (K74.3).

### Data collection

All of these patients were then assessed for the presence of several cardiovascular co-morbidities and risk factors, including: congestive heart failure (CHF, I150.0), atrial fibrillation (I48), and non-insulin dependent diabetes mellitus (NIDDM, E11), chronic kidney disease (N18), obesity (E66.0), myocardial infarction (I21-I22), ischaemic heart disease (I20-25), ischaemic stroke (I63.9), hyperlipidaemia (E78.5), hypertension (I10), and peripheral vascular disease (I73.9). Prevalence of malignancy was collected using codes C0-C9. Patients were also assessed for liver-related events: hepatocellular carcinoma (HCC, C22.9), hepatic failure (K72), oesophageal varices (I85), portal hypertension (K76.6), splenomegaly (R16.1), and ascites (R18). A combined 'hepatic decompensation or failure' score was generated from the sum of all non-malignant liver-related events. Inclusion of hepatic encephalopathy (K72.9) or variceal

bleeding (I98.3, I98.8, I85.9) did not identify any additional patients. Jaundice was not included due to identification of patients with obstructive (non-hepatic) jaundice.

Vital status (alive or deceased) on 31st March 2013 was determined by record linkage to the National Health Tracing Services (NHS strategic tracing service) and was received along with the raw data; this was used to calculate all-cause mortality and survival. The first admission to hospital treatment was chosen as index admission, follow-up of patients continued until 31st March 2013.

### Approvals

Confidentiality of information was maintained in accordance with the UK Data Protection Act. The patient data included was fully anonymous and non-identifiable when received by the authors, and collected routinely by the hospitals. Therefore, according to local research ethics policies we were not required to seek formal ethical approval for this study.

### Statistical analysis

Data analysis was performed using SPSS version 20.0 (SPSS Inc. Chicago, IL), R version 4.0.2, and GraphPad Prism version 8.0. Code used in analyses in R is available in the S1 File. Analyses were initially performed by comparing three groups: control, non-cirrhotic NAFLD (NAFL plus NASH), and NAFLD-cirrhosis. Clinical outcomes were compared between groups using chi-squared tests for categorical variables and t-tests for age. Multivariate logistic regression was used to determine adjusted odds ratios for hepatic failure/decompensation and hepatocellular carcinoma, adjusted for demographics (age, gender, ethnicity). Cox regression analysis was used to determine adjusted hazard ratios (HR) for overall mortality between groups. Cox regression was performed using four models: 1) adjusting for variations in demographics (age, gender, ethnicity); 2) adjusting for demographics plus CVD or metabolic risk factors (obesity, NIDDM, CHF, ischaemic stroke, myocardial infarction, chronic kidney disease, peripheral vascular disease, hypertension, hyperlipidaemia, ischaemic heart disease, and atrial fibrillation); 3) adjusting for demographics plus liver-related events (hepatocellular carcinoma, hepatic failure, oesophageal varices, portal hypertension, splenomegaly, and ascites); and 4) adjusting for demographics, CVD, metabolic risk factors, and liver-related events. Participants with incomplete data were excluded. Cumulative hazard survival curves were derived using Cox proportional regression models. A supplementary analysis was performed using four groups, by splitting the non-cirrhotic NAFLD group into NAFL and NASH: control, NAFL, NASH, and NAFLD-cirrhosis. All analyses from the main analyses were replicated for these four groups.

A further sub-analysis was performed by removing all participants with liver-related events and then calculating adjusted mortality hazard ratios using Cox regression adjusted for 1) demographics only, and 2) demographics, CVD, and metabolic risk factors.

Two-sided p-values were calculated for all statistical tests and then corrected for multiple testing using the Benjamini-Hochberg method. Q-values <0.05 were considered significant.

## Results

1,802 patients were identified with NAFLD-spectrum diagnoses, of which 1,091 had non-cirrhotic NAFLD (994 with NAFL and 97 with NASH) and 711 with NAFLD-cirrhosis (Table 1 and S1 Table). They were matched to 24,737 hospitalised in-patient controls. The median duration of follow-up for each group was: control 5.3 years, NAFL 4.6 years, NASH 4.4 years, and cirrhosis 2.8 years (range 1 day—14 years for all groups).

**Table 1. Demographics for control, non-cirrhotic-NAFLD and NAFLD-cirrhosis patients.**

|  | Control (n = 24,737) | NAFLD (n = 1,091) | NAFLD vs. Control q-value | Cirrhosis (n = 711) | Cirrhosis vs. Control q-value | Cirrhosis vs. NAFLD q-value |
|---|---|---|---|---|---|---|
| Mean age (SD) | 55.0 (15.4) | 51.6 (15.2) | 6.4E-12 | 63.6 (13.5) | 2.0E-53 | 2.0E-62 |
| Female | 10,999 (44.5) | 518 (47.5) | 0.09 | 338 (47.5) | 0.15 | 1 |
| Caucasian | 19,761 (79.9) | 850 (77.9) | 0.17 | 607 (85.4) | 6.5E-04 | 2.10E-04 |
| South Asian | 1,652 (6.7) | 140 (12.8) | 4.6E-14 | 37 (5.2) | 0.17 | 4.70E-07 |

Baseline demographics for participants included in the study. Q-values were derived from chi-squared tests (for sex and ethnicity) or t-tests (for age) with adjustment for multiple testing using the Benjamini-Hochberg method. SD, standard deviation.

The control group was older than the non-cirrhotic NAFLD group (55 years vs. 52 years, q = $6.4 \times 10^{-12}$) but younger than the cirrhosis group (55 years vs. 64 years, q = $2.0 \times 10^{-62}$). The non-cirrhotic NAFLD group had a higher proportion of participants self-identifying as South Asian (6.6% control vs. 12.8% non-cirrhotic NAFLD, q = $4.6 \times 10^{-14}$).

Patients with non-cirrhotic NAFLD had a higher prevalence of metabolic risk factors (hyperlipidaemia, type 2 diabetes mellitus, obesity, and hypertension) than controls (Table 2), despite being younger. There were no differences between NAFL and NASH groups on sub-analysis (S2 Table). Compared to the NAFLD group, patients with cirrhosis were more likely to have T2DM (22% NAFLD vs. 35% cirrhosis, q = $8.7 \times 10^{-10}$) but had a lower prevalence of hyperlipidaemia (13.3% NAFLD vs. 6.5% cirrhosis, q = $1.4 \times 10^{-5}$).

There was an increasing burden of cardiovascular co-morbidity with more advanced liver disease. Compared to the control group, patients with NASH had a higher prevalence of heart failure (10.5% vs. 3.5%, q = $4.5 \times 10^{-3}$, S2 Table). The cirrhosis group showed higher prevalence of atrial fibrillation and CKD compared to the NAFLD group (Table 2). No differences in prevalence of ischaemic stroke were observed.

**Table 2. Cardiovascular disease burden and liver-related events across the NAFLD spectrum.**

|  | Control (n = 24,737) | NAFLD (n = 1,091) | NAFLD vs. Control q-value | Cirrhosis (n = 711) | Cirrhosis vs. Control q-value | Cirrhosis vs. NAFLD q-value |
|---|---|---|---|---|---|---|
| Obesity | 307 (1.2) | 92 (8.4) | 3.6E-77 | 27 (3.8) | 2.0E-08 | 2.8E-04 |
| Type 2 Diabetes | 2328 (9.4) | 235 (21.5) | 4.2E-38 | 250 (35.2) | 3.3E-110 | 8.7E-10 |
| Hyperlipidaemia | 2015 (8.1) | 145 (13.3) | 7.6E-09 | 46 (6.5) | 0.16 | 1.4E-05 |
| Hypertension | 5655 (22.9) | 343 (31.4) | 2.2E-10 | 225 (31.6) | 1.1E-07 | 1 |
| Ischaemic heart disease | 2951 (11.9) | 115 (10.5) | 0.24 | 112 (15.8) | 4.0E-03 | 2.2E-03 |
| Myocardial infarction | 940 (3.8) | 24 (2.2) | 0.02 | 21 (3.0) | 0.33 | 0.48 |
| Atrial fibrillation | 1174 (4.7) | 64 (5.9) | 0.16 | 86 (12.1) | 3.3E-18 | 1.1E-05 |
| Congestive Heart Failure | 865 (3.5) | 44 (4.0) | 0.45 | 63 (8.9) | 2.7E-13 | 7.4E-05 |
| Ischaemic stroke | 498 (2.0) | 12 (1.1) | 0.08 | 24 (3.4) | 0.03 | 2.2E-03 |
| Peripheral vascular disease | 381 (1.5) | 13 (1.2) | 0.47 | 14 (2.0) | 0.49 | 0.33 |
| Chronic Kidney Disease | 309 (1.2) | 23 (2.1) | 0.04 | 33 (4.6) | 8.8E-14 | 5.5E-03 |
| Any malignancy | 1876 (7.6) | 103 (9.4) | 0.05 | 176 (24.8) | 1.2E-60 | 1.1E-17 |
| GI malignancy | 395 (1.6) | 38 (3.5) | 9.4E-06 | 111 (15.6) | 3.4E-151 | 6.6E-19 |
| Hepatic failure/ decompensation | 116 (0.5) | 61 (5.6) | 1.3E-86 | 300 (42.2) | <1E-300 | 1.0E-78 |
| Hepatocellular carcinoma | 45 (0.2) | 14 (1.3) | 4.5E-12 | 93 (13.1) | <1E-300 | 6.5E-24 |
| All-cause mortality | 3,635 (14.7) | 149 (13.7) | 0.44 | 288 (40.5) | 1.9E-225 | 5.9E-91 |

Crude rates of mortality, metabolic, cardiovascular, and liver-related outcomes for control, non-cirrhotic-NAFLD, and NAFLD-cirrhosis, patients during a 14-year study period. Q-values were derived from chi-squared tests with adjustment for multiple testing using the Benjamini-Hochberg method.

The prevalence of hepatic events (hepatic failure or decompensation and development of hepatocellular carcinoma (HCC)) was higher in all groups of patients with NAFLD (Table 2) and was similar between NAFL and NASH groups (S2 Table). The increased prevalence of liver-related events remained after adjustment for age, sex, and ethnicity (Table 3).

Unadjusted 14-year all-cause mortality was 14.7% for patients in the control group, 13.7% for patients with non-cirrhotic NAFLD, and 40.5% for those with NAFLD-cirrhosis (Table 2). However, the control group were significantly older than those with non-cirrhotic NAFLD, therefore after adjustment for age, gender and ethnicity, all-cause mortality hazard ratio was higher in the non-cirrhotic NAFLD group compared to the control group (HR 1.3 (95% CI 1.1–1.5), q = $3.7 \times 10^{-3}$, Table 3 and Fig 1A). After adjustment for cardiovascular factors, all-cause mortality was still elevated compared to the control group: non-cirrhotic NAFLD HR 1.2 (95% CI 1.0–1.4, q = 0.04, Fig 1B) and NAFLD-cirrhosis HR 3.4 (95% CI 2.8–4.2, q = $1.1 \times 10^{-33}$). However, after adjusting for liver-related events there was no difference in mortality between controls and non-cirrhotic NAFLD patients (HR 1.1 (95% CI 0.9–1.3), q = 0.61, Fig 1C) but elevated mortality remained for those with NAFLD-cirrhosis (HR 3.0 (95% CI 2.4–3.7), q = $4.8 \times 10^{-24}$). No differences in adjusted mortality ratios were observed between NAFL and NASH groups on sub-analysis (S1 Fig and S3 Table).

After removing patients who had experienced liver-related events (S4 Table), there was no difference in hazard ratio of mortality for patients with non-cirrhotic NAFLD compared to control when adjusting for demographics (HR 1.2 (95% CI 1.0–1.4), q = 0.08).

## Discussion

This study provides important non-specialist "real life" data amongst hospitalised patients demonstrating an increased mortality for patients with non-cirrhotic NAFLD, even after adjustment for CVD. This difference appeared to be due to liver-related events (hepatic

**Table 3. Adjusted odds ratios for liver-related outcomes and adjusted mortality hazard ratios.**

| | NAFLD vs. Control | | Cirrhosis vs. Control | | Cirrhosis vs. NAFLD | |
|---|---|---|---|---|---|---|
| | Adj. OR (95% CI) | q-value | Adj. OR (95% CI) | q-value | Adj. OR (95% CI) | q-value |
| Hepatic failure/decompensation | 2.6 (2.3–2.9) | 4.00E-57 | 4.9 (4.7–5.2) | <1E-300 | 2.4 (2.1–2.7) | 3.60E-49 |
| Hepatocellular carcinoma | 2.2 (1.5–2.7) | 1.10E-11 | 4.2 (3.8–4.6) | 8.30E-107 | 1.9 (1.4–2.6) | 7.60E-11 |
| | HR (95% CI) | q-value | HR (95% CI) | q-value | HR (95% CI) | q-value |
| Mortality adjusted for demographic characteristics | 1.3 (1.1–1.5) | 3.70E-03 | 3.8 (3.4–4.2) | 8.30E-146 | 3.7 (3.0–4.5) | 1.30E-38 |
| Mortality adjusted for demographics, metabolic risk factors, and CVD | 1.2 (1.0–1.4) | 0.04 | 3.2 (2.9–3.6) | 8.30E-107 | 3.4 (2.8–4.2) | 1.10E-33 |
| Mortality adjusted for demographics and liver-related events | 1.1 (0.9–1.3) | 0.61 | 2.4 (2.1–2.8) | 2.80E-31 | 3.0 (2.4–3.7) | 4.80E-24 |
| Mortality adjusted for demographics, metabolic risk factors, CVD, and liver-related events | 1.0 (0.9–1.2) | 0.8 | 2.1 (1.8–2.4) | 9.00E-23 | 2.8 (2.3–3.5) | 2.00E-21 |

Odds ratios for liver-related events (hepatic failure/decompensation and hepatocellular carcinoma) were calculated using multivariable logistic regression adjusted for age, sex, and ethnicity. Adjusted hazard ratios of overall mortality were calculated between control, non-cirrhotic NAFLD, and NAFLD-cirrhosis groups using Cox proportional regression. Adjustment for demographic characteristics were gender, age and ethnicity. Adjustment for metabolic risk factors and CVD included: obesity, type 2 diabetes mellitus, CHF, ischaemic stroke, myocardial infarction, chronic kidney disease, peripheral vascular disease, hypertension, hyperlipidaemia, ischaemic heart disease, and atrial fibrillation. Adjustment for liver-related events included: hepatocellular carcinoma, hepatic failure, oesophageal varices, portal hypertension, splenomegaly, and ascites. Control n = 25,780; NAFLD n = 1,343; and Cirrhosis n = 1,235. Q-values were calculated from p-values using the Benjamini-Hochberg method. Adj. OR, adjusted odds ratio; CI, confidence interval; CVD, cardiovascular disease; HR, hazard ratio.

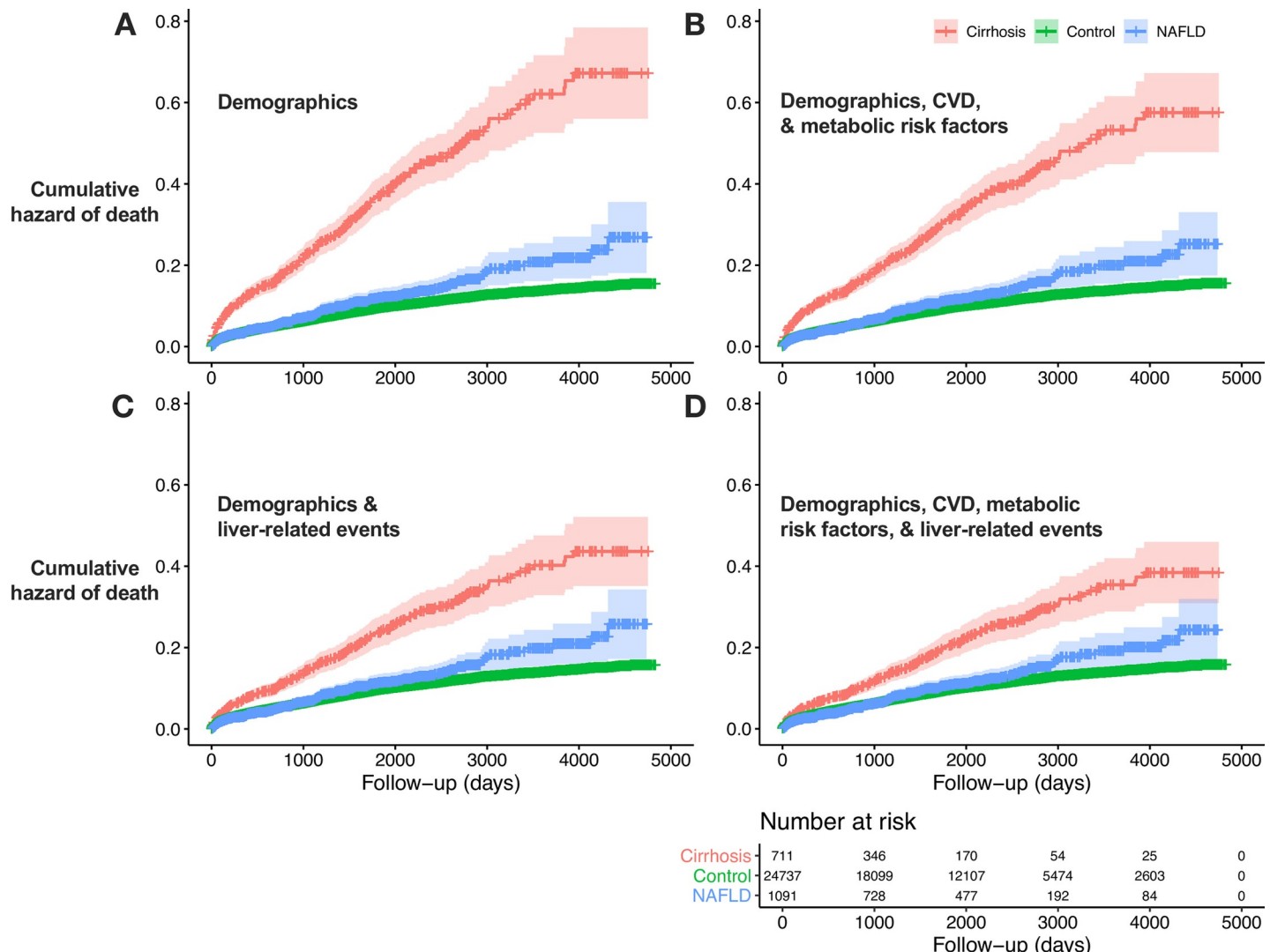

**Fig 1. Adjusted cumulative hazard of mortality for hospitalised controls, patients with non-cirrhotic NAFLD, and NAFLD-cirrhosis.** Survival curves showing cumulative hazard of mortality derived from four models of adjustment using Cox proportional regression. Data shows 95% CI for control n = 24,737; NAFLD n = 1,091; and Cirrhosis n = 711. (A) Adjustment for demographic characteristics only (gender, age and ethnicity). (B) Adjustment for demographics plus CVD and metabolic risk factors (obesity, type 2 diabetes mellitus, CHF, ischaemic stroke, myocardial infarction, chronic kidney disease, peripheral vascular disease, hypertension, hyperlipidaemia, ischaemic heart disease, and atrial fibrillation). (C) Adjustment for demographics and liver-related events (hepatocellular carcinoma, hepatic failure, oesophageal varices, portal hypertension, splenomegaly, and ascites). (D) Adjustment for demographics, CVD, metabolic risk factors, and liver-related events.

failure/decompensation and HCC) over a 14 year follow-up period. The size of the cohort and lack of link to specialist liver centres reduces likelihood of bias. These data will help inform healthcare demand for this cohort of patients, complementing modelling estimates [31, 32]. In addition, we have highlighted the burden of CVD in patients with NAFLD-cirrhosis, which poses a particular issue for transplantation. Whilst all participants in our cohort were hospital-ised, which may increase their risk of future clinical events, so were the controls, thus the com-parisons between groups remain valid within this setting.

The strong association between NAFLD and CVD has been well established [33–35]. Rela-tionships have been identified between NAFLD and heart failure [36], atrial fibrillation [37], hypertension [38], stroke [39], chronic kidney disease [40], and coronary artery disease. NAFLD has even been linked to increased mortality in acute heart failure [41]. However

strong observational data from a large European meta-analysis suggests that NAFLD is not causal in acceleration of CVD [13]. Our data highlights the particularly increased prevalence in patients with cirrhosis and is broadly consistent with a previously reported cohort from Italy [19]. Indeed, hypertension has been highlighted as an independent risk factor for advanced fibrosis in NAFLD [42]. A further consideration is whether heart failure contributes to accelerated fibrosis in NAFLD [43], though causality is difficult to establish.

Some previous studies with biopsy-defined cohorts have been smaller, did not adjust for cardiovascular diagnoses [3, 22], and found no difference in mortality between participants with NAFLD and no fibrosis, and controls. Kim *et al.* used NHANES data to stratify patients by non-invasive fibrosis scores, and again found no increase in mortality in patients with ultra-sound-defined NAFLD, after correction for diabetes and hypertension [21]. This may be accounted for by differences in the ethnicity of the cohort and also the general, rather than specialist, nature of our population. A more recent Italian study in a prospective, consecutively recruited cohort found that after adjusting for metabolic risk factors patients with NAFLD had higher rates of cardiovascular events [10]. Moreover, they used non-invasive fibrosis scores to demonstrate that higher fibrosis stage was associated with increased risk of cardiovascular events, which is generally consistent with our results from the NAFLD-cirrhosis group. Similar results were observed in a further biopsy-proven NAFLD cohort of 285 patients [44].

NAFLD may itself be a marker of sub-clinical CVD. For example, increased carotid intima media thickness has been found in adolescents with NAFLD [45]. This is mechanistically plausible as hepatic steatosis occurs (in part) secondary to peripheral insulin resistance and elevated substrate delivery from lipolysis of adipose tissue. Steatosis itself then contributes to systemic insulin resistance [14, 46]. Therefore, in this analysis, despite adjusting for metabolic covariates and cardiovascular risk factors, elevated mortality may reflect the sub-clinical nature of atherosclerosis associated with NAFLD, even at an early stage.

Given the shared disease mechanisms and clinical outcomes for NAFLD and CVD, these data suggest a common framework for treatment. Weight loss is the only established treatment strategy for NAFLD [47] and there is data suggesting that specific dietary regimens (including the Mediterranean diet) are beneficial [48]. The same lifestyle interventions and aggressive risk factor modification will have dual impact on reducing cardiovascular [49, 50] and hepatic events. Statins are likely to be recommended for many patients with NAFLD due to their CVD risk factors, often in combination with treatment for type 2 diabetes. Given the relative risk of incident morbidity from CVD compared to liver-related events, these treatments are likely to primarily influence cardiac events [51, 52]. We were unable to determine drug therapy in the participants included in this study, which has the potential to influence disease outcomes [53, 54] and therefore should be considered as a limitation of the work.

Whilst we were not able to determine cause of death or admission in our cohort, we were able to determine that liver decompensation events were increased in all groups relative to the control group. In addition, excluding participants who had experienced liver-related events, removed the mortality difference between control and non-cirrhotic NAFLD patients.

This study is limited by its retrospective design and the use of generic coding, which did not provide information on how the diagnosis was obtained i.e. imaging, liver function tests, or liver biopsy and we were unable to identify whether NAFLD was the cause of admission or an existing co-morbidity in cases. Therefore, the sub-analysis for NAFL and NASH groups should be interpreted with this in mind. However, coding improvements along with standardised diagnosis of NAFLD in the UK means that the impact of inaccurate coding may be low.

ICD-10 coding is likely to be a poorly sensitive method to exclude NAFLD from the control group, due to the asymptomatic nature of the condition. However this bias would likely result in an under-estimation of the hazard ratios comparing control and NAFLD groups. The use of

ICD-10 codes for the exclusion of other causes of liver dysfunction (for example, viral hepatitis) and patients with a history of alcohol consumption may have also contributed to inaccurate coding. However, such biases have been limited from our previous study looking at the association between cardiovascular and respiratory conditions [29]. There is likely to be a degree of under coding of NAFLD, especially as clinical awareness of NAFLD was not optimal at the beginning of the data capture [55]. It should be noted that the follow-up in this study ended in 2013, and since there has been a progressive increase in prevalence of NAFLD as well as development in non-invasive diagnostic techniques that were not widely available in 2000–2013 (e.g. elastography)

Our analysis has illustrated that a substantial proportion of participants with cirrhosis have risk factors for vascular events. Cirrhosis is a state of disordered clotting with patients at increased risk for thrombosis yet also exhibit hyperfibrinolysis [56]. In this study we were unable to further examine the characteristics of participants experiencing vascular events due to a lack of detailed participant-level data.

Whilst the participants with non-cirrhotic NAFLD in this study were from non-specialist centres, they may still represent a more advanced subset of patients with NAFLD than the entire population of individuals with hepatic steatosis. We found 5.6% of patients initially diagnosed with non-cirrhotic NAFLD to experience a liver-related event during 14-years of follow-up, compared to 7.9% found in the biopsy-staged cohort from Hagström et al. [7] and 4.2% reported by Angulo et al. [3] The control cohort used in this study were also hospitalised patients, therefore comparisons using hazard ratios are valid within this setting. However our results are not generalisable to patients diagnosed with NAFLD outside of a hospital setting.

Similarly, the results comparing non-cirrhotic NAFLD with NAFLD-cirrhosis are likely to have been influenced by lead time bias. The cirrhosis group were significantly older than non-cirrhotic NAFLD group and were, by definition, at a more advanced stage in their disease. Therefore, the NAFLD-cirrhosis results illustrate a highly concentrated group of the most severe part of the NAFLD spectrum.

In conclusion, these results contribute to our understanding of co-morbidity, mortality and liver decompensation in patients hospitalised with NAFLD spectrum disease and demonstrates there is higher mortality independent of known cardiovascular risk factors.

## Supporting information

**S1 Table. Demographics for control, NAFL, NASH and NAFLD-cirrhosis patients.** Baseline demographics for participants included in the study. Q-values were derived from chi-squared tests (for sex and ethnicity) or t-tests (for age) with adjustment for multiple testing using the Benjamini-Hochberg method. SD, standard deviation.
(DOCX)

**S2 Table. Cardiovascular disease burden and liver-related events across the NAFLD spectrum.** Crude rates of mortality, metabolic, cardiovascular, and liver-related outcomes for control, non-cirrhotic-NAFLD, and NAFLD-cirrhosis, patients during a 14-year study period. Data is given a number of events (%). Q-values were derived from chi-squared tests with adjustment for multiple testing using the Benjamini-Hochberg method. GI malignancy, gastrointestinal malignancy.
(DOCX)

**S3 Table. Adjusted odds ratios for liver-related outcomes and adjusted mortality ratios.** Odds ratios for liver-related events (hepatic failure/decompensation and hepatocellular carcinoma) were calculated using multivariable logistic regression adjusted for age, sex, and

ethnicity. Adjusted hazard ratios of overall mortality were calculated between control, non-cirrhotic NAFLD, and NAFLD-cirrhosis groups using Cox proportional regression. Adjustment for demographic characteristics were gender, age and ethnicity. Adjustment for cardiovascular risk factors and CVD included: obesity, type 2 diabetes mellitus, CHF, ischaemic stroke, myocardial infarction, chronic kidney disease, peripheral vascular disease, hypertension, hyperlipidaemia, ischaemic heart disease, and atrial fibrillation. Adjustment for liver-related events included: hepatocellular carcinoma, hepatic failure, oesophageal varices, portal hypertension, splenomegaly, and ascites. Control n = 25,780; NAFL n = 1,238; NASH n = 105; and Cirrhosis n = 1,235 Q-values were calculated from p-values using the Benjamini-Hochberg method. Adj. OR, adjusted odds ratio; CI, confidence interval; CVD, cardiovascular disease; HR, hazard ratio.
(DOCX)

**S4 Table. Adjusted mortality ratios for patients with no liver-related events.** Adjusted hazard ratios of overall mortality were calculated between control, non-cirrhotic NAFLD, and NAFLD-cirrhosis groups using Cox proportional regression after removal of all participants with liver-related events. Adjustment for demographic characteristics were gender, age and ethnicity. Adjustment for cardiovascular risk factors and CVD included: obesity, type 2 diabetes mellitus, CHF, ischaemic stroke, myocardial infarction, chronic kidney disease, peripheral vascular disease, hypertension, hyperlipidaemia, ischaemic heart disease, and atrial fibrillation. Control n = 25,551, NAFLD n = 1,241, and cirrhosis n = 591. Q-values were calculated from p-values using the Benjamini-Hochberg method. CI, confidence interval; CVD, cardiovascular disease; HR, hazard ratio.
(DOCX)

**S1 Fig. Adjusted cumulative hazard of mortality for hospitalised controls, patients with NAFL, NASH, and NAFLD-cirrhosis.** Survival curves showing cumulative hazard of mortality derived from four models of adjustment using Cox proportional regression. Data shows 95% CI for control n = 24,737; NAFL n = 994, NASH n = 97; and Cirrhosis n = 711. (A) adjustment for demographic characteristics only (gender, age and ethnicity). (B) adjustment for demographics plus CVD and metabolic risk factors (obesity, type 2 diabetes mellitus, CHF, ischaemic stroke, myocardial infarction, chronic kidney disease, peripheral vascular disease, hypertension, hyperlipidaemia, ischaemic heart disease, and atrial fibrillation). (C) adjustment for demographics and liver-related events (hepatocellular carcinoma, hepatic failure, oesophageal varices, portal hypertension, splenomegaly, and ascites). (D) adjustment for demographics, CVD, metabolic risk factors, and liver-related events.
(DOCX)

**S1 File. Code used in analyses in R 4.0.**
(DOCX)

## Author Contributions

**Conceptualization:** Jake P. Mann, Matthew J. Armstrong, Philip N. Newsome, Rahul Potluri.

**Data curation:** Paul Carter, Hesham K. Abdelaziz, Hardeep Uppal, Billal Patel, Suresh Chandran, Ranjit More, Rahul Potluri.

**Formal analysis:** Jake P. Mann, Paul Carter.

**Project administration:** Hesham K. Abdelaziz, Hardeep Uppal, Billal Patel, Suresh Chandran, Ranjit More, Philip N. Newsome, Rahul Potluri.

**Resources:** Hesham K. Abdelaziz, Hardeep Uppal, Billal Patel, Suresh Chandran, Ranjit More.

**Supervision:** Philip N. Newsome, Rahul Potluri.

**Writing – original draft:** Jake P. Mann.

**Writing – review & editing:** Jake P. Mann, Paul Carter, Matthew J. Armstrong, Hesham K. Abdelaziz, Hardeep Uppal, Billal Patel, Suresh Chandran, Ranjit More, Philip N. Newsome, Rahul Potluri.

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
