## [Decision Letter · Decision Letter 0]

28 Jul 2020

PONE-D-19-32801

Hospital admission with non-alcoholic fatty liver disease is associated with increased all-cause mortality independent of cardiovascular risk factors

PLOS ONE

Dear Dr. Mann,

Thank you for submitting your manuscript to PLOS ONE. I sincerely apologise for the unusually delayed review timeframe for your manuscript. After careful consideration, we feel that it has merit but does not fully meet PLOS ONE’s publication criteria as it currently stands. Therefore, we invite you to submit a revised version of the manuscript that addresses the points raised during the review process.

Your manuscript has been assessed by four reviewers, whose comments are appended below. Although they are all generally quite positive about the study, they raise some concerns about the study timeframe, definition of study populations, and statistical analysis that should be discussed or addressed with the appropriate revisions. In addition, please feel free to use your discretion when deciding whether to include additional references in your revised manuscript.

We look forward to receiving your revised manuscript.

Kind regards,

Emily Chenette

Deputy Editor in Chief

PLOS ONE

Journal Requirements:

Additional Editor Comments (if provided):

Reviewers' comments:

Reviewer's Responses to Questions

**Comments to the Author**

1. Is the manuscript technically sound, and do the data support the conclusions?

Reviewer #1: Yes

Reviewer #2: Yes

Reviewer #3: Yes

Reviewer #4: Partly

2. Has the statistical analysis been performed appropriately and rigorously? 

Reviewer #1: Yes

Reviewer #2: Yes

Reviewer #3: Yes

Reviewer #4: No

3. Have the authors made all data underlying the findings in their manuscript fully available?

Reviewer #1: Yes

Reviewer #2: Yes

Reviewer #3: Yes

Reviewer #4: Yes

4. Is the manuscript presented in an intelligible fashion and written in standard English?

Reviewer #1: Yes

Reviewer #2: Yes

Reviewer #3: Yes

Reviewer #4: Yes

5. Review Comments to the Author

Reviewer #1: The research paper PONE-D-19-32801 entitled “Hospital admission with non-alcoholic fatty liver disease is associated with increased all-cause mortality independent of cardiovascular risk factors” has relevance to the scope and the audience of PONE.

This paper aimed to assess the mortality in NAFLD when adjusting for CVD risk factors in a ‘real world’ cohort of inpatients.

The authors suggest that there is a high burden of cardiovascular disease in NAFLD-cirrhosis patients. From a large “real-life” non-specialist registry of hospitalized patients, NAFLD patients have increased overall mortality and rate of liver-related complications compared to controls after adjusting for cardiovascular disease.

Major comments for the authors

1. This analysis was performed on an interesting issue: The effect of NAFLD on overall mortality in a real world setting regardless of other CVD risk factors.

2. It is a well written paper. The text, the figure and the tables are of appropriate extent and content as well as informative.

3. The references are up to date.

4. The results of the paper have practical implications in patients with NAFLD, especially those with cirrhosis.

Reviewer #2: This is an interesting study based on “real world” data.

1] The authors need to be more careful in using abbreviations. For example, “cardiovascular disease” is abbreviated in the Abstract but not used throughout the Abstract. Also, this term is not abbreviated the first time it appears in the Introduction.

The same for NAFLD, HCC etc.

Please check the whole text for similar errors.

2] Was p value 2-sided?

3] Useful refs to comment:

Athyros VG, Tziomalos K, Katsiki N, Doumas M, Karagiannis A, Mikhailidis DP. Cardiovascular risk across the histological spectrum and the clinical manifestations of non-alcoholic fatty liver disease: An update. World J Gastroenterol. 2015;21(22):6820‐6834.

Le MH, Devaki P, Ha NB, et al. Prevalence of non-alcoholic fatty liver disease and risk factors for advanced fibrosis and mortality in the United States. PLoS One. 2017;12(3):e0173499.

4] The median duration of follow-up differed between groups. Does this translate to the severity of the disease in each group? For example, do patients with cirrhosis died earlier? Please comment.

5] Although lifestyle interventions do remain the first-line therapeutic option for NAFLD/NASH, other drugs may be helpful and especially statins and antidiabetic drugs. A brief comment should be added. Useful refs

Athyros VG, Alexandrides TK, Bilianou H, et al. The use of statins alone, or in combination with pioglitazone and other drugs, for the treatment of non-alcoholic fatty liver disease/non-alcoholic steatohepatitis and related cardiovascular risk. An Expert Panel Statement. Metabolism. 2017;71:17‐32.

Stahl EP, Dhindsa DS, Lee SK, Sandesara PB, Chalasani NP, Sperling LS. Nonalcoholic Fatty Liver Disease and the Heart: JACC State-of-the-Art Review. J Am Coll Cardiol. 2019;73(8):948‐963.

6] Drug therapy may affect clinical outcomes. The inability to evaluate drug effects in this study should be mentioned in the limitations.

Katsiki N, Perakakis N, Mantzoros C. Effects of sodium-glucose co-transporter-2 (SGLT2) inhibitors on non-alcoholic fatty liver disease/non-alcoholic steatohepatitis: Ex quo et quo vadimus?. Metabolism. 2019;98:iii‐ix.

Athyros VG, Polyzos SA, Kountouras J, et al. Non-Alcoholic Fatty Liver Disease Treatment in Patients with Type 2 Diabetes Mellitus; New Kids on the Block. Curr Vasc Pharmacol. 2020;18(2):172‐181.

Reviewer #3: This is a retrospective study based on a registry data. I have some comments and suggestions.

Why the inclusion period was only until to 2013 (and not until 2018-2019) and why now do you choose to publish these data?

The diagnosis of this pathology have suffered modification between 2000 and 2013. Therefore, the codification is not sufficiently to be sure that the patients was appropriately included. What where the modalities of diagnosis for these pathologies? I think that this is the main cornerstone of this study (the inhomogeneity of diagnosis for these pathologies).

In table 1, I do not see the statistical significance (p). It would be important. All-cause mortality has the same percentage in controls and NAFL.

In table 2 why do you not put the sixth column for p? It would be easier to follow these data.

``The cirrhosis group showed higher prevalence of heart failure, atrial fibrillation, CKD, and ischaemic heart disease, compared to the NAFL group.`` I suggest to analyze separately this subgroup of patients. It might offers very interestingly data about the relationship between thrombosis risks in patients with cirrhosis.

Reviewer #4: In “Hospital admission with non-alcoholic fatty liver disease is associated with increased all-cause mortality independent of cardiovascular risk factors” Mann and collegues tried to define whether NAFL and NASH are associated with increased all-cause mortality in hospitalised patients. The study was conducted as a retrospective cohort study using ICD-10 coding to define the presence of NAFL, NASH and NAFLD related cirrhosis.

Output of the study were: overall mortality and hepatic outcomes (liver decompensation and HCC). In addition, authors characterized metabolic and cardiovascular comorbidities of NAFLD patients recording specific ICD code, listed in methods section.

Major:

I have some concerns about the populations definition, the statistical methods and the strength of the results.

Populations definition:

1. As authors reported in study limitation section, the use of ICD code to define the presence or the absence of NAFL/NASH, is highly defective. However, some bias could be reduced considering NAFL and NASH as a unique non cirrhotic NAFLD group. Often, asymptomatic NAFLD patients classified as NAFL patients hide NASH patients with normal liver function tests and non-invasive markers of fibrosis.

2. The absence of an ICD10 which refers to NAFLD spectrum does not allow to exclude the presence of nafld among controls (as stated in study limitation section). However, the authors should stress that this bias could underestimated the HR in the analysis and not conversely.

Statistical methods:

3. Since the authors talk about prevalence of liver decompensation/failure or HCC, I suppose that were recorded ad admission in hospital (follow up start time). In statistical methods authors properly say they used Multivariate logistic regression to define factors associated with this outcome. However, they improperly use the “hazard ratio” to define the statistical outcome. In this case, being a no time-corrected method, the statistical outcome is “odds ratio”, please amend properly

4. Figure 1. I think that the Kaplan Mayer curves, being the graphic representation of an univariate model, are not the best graphic model to represent survival in this study. At follow up start time patients and controls had different prevalence of cardiovascular (and liver-related?) conditions that may affect survival. Please consider using multiple survival curve (controls vs NAFLD (as suggested in point 1) and control vs patients with cirrhosis) derived from cox-regression analyses.

5. Why liver decompensation/failure/HCC was not included in cox multivariate regression? The differences in survival may partially been explained by different prevalence of liver complications. An interesting way to test this hypothesis could be an additional cox regression analysis performed after excluding patients whit liver complications

Strength of results:

6. Concern reported in point 5 and the inability to define with certain the cause of hospital admission (follow up start time) raise some doubts that NAFLD patients included in this cohort could represents a cluster of most severe patients in which NAFLD plays a key role.

Minor:

1. Lines 124: another study described the CV burden in patients with NAFLD-cirrhosis, please discuss it (https://doi.org/10.1016/j.atherosclerosis.2017.03.038)

2. Table 1: Adjusted and unadjusted HRs for overall mortality and ORs (?) for Liver failure/decompensation and HCC should be reported in a separate table.

3. The sentence “increased mortality for patients with NAFLD, irrespective of fibrosis” (lines 277-278) is not supported by results. In this study, the only differentiation in the severity of fibrosis could be made comparing cirrhotic with non-cirrhotic patients and, among them there is a wide difference in survival (HR 3.8 vs 2.6 vs. 65.4!)

4. Lines 298-304: a recent study proved higher incidence of CV disease in patients with NAFLD compared to metabolic controls. In addition, in the same study, non-invasive markers of fibrosis identify patients at higher risk for CV events among those with NAFLD. The same study could be discussed in introduction section (lines 101-107) to affirm that fibrosis, in NAFLD patients, is the major predictors of CVE also. doi: 10.1016/j.cgh.2019.12.026

5. Please check through the paper the correct use of NAFLD/NAFL terms. Eg: in figure 1 “NAFLD” was incorrectly used instead of NAFL

6. PLOS authors have the option to publish the peer review history of their article (what does this mean?). If published, this will include your full peer review and any attached files.

Reviewer #1: No

Reviewer #2: No

Reviewer #3: No

Reviewer #4: No

---

## [Author Response · Author response to Decision Letter 0]

14 Sep 2020

Thank you for your review of our manuscript. We have carefully responded to the Reviewers’ comments and feel that we have been able to address them all adequately, such that our manuscript is substantially improved. The major changes we have made are:

– Combining NAFL and NASH groups into a single ‘non-cirrhotic NAFLD’ group

– Inclusion of liver-related events as co-variates in Cox regression analyses

– Sub-analysis by removing patients with liver-related events

– Re-formatting and re-arranging our results tables as suggested by the Reviewers

– Reviewed the inclusion/exclusion criteria for all patients in light of recent guidelines and comments from the Reviewers

– Changing our figure to a cumulative hazard survival curve derived from the Cox proportional regression models

Below is a point-by-point responses to the Reviewers’ comments.

[Line numbers refer to the ‘clean’ manuscript version.]

Reviewer #1:

Thank you for your comments. We have updated our manuscript in response to some suggestions from the other Reviewers.

Reviewer #2: This is an interesting study based on “real world” data.

1] The authors need to be more careful in using abbreviations. For example, “cardiovascular disease” is abbreviated in the Abstract but not used throughout the Abstract. Also, this term is not abbreviated the first time it appears in the Introduction.

The same for NAFLD, HCC etc.

Please check the whole text for similar errors.

Apologies for these inconsistencies. We have carefully corrected our abbreviations throughout.

2] Was p value 2-sided?

Yes, we have now added this into the Methods (line 213).

3] Useful refs to comment:

Athyros VG, Tziomalos K, Katsiki N, Doumas M, Karagiannis A, Mikhailidis DP. Cardiovascular risk across the histological spectrum and the clinical manifestations of non-alcoholic fatty liver disease: An update. World J Gastroenterol. 2015;21(22):6820‐6834.

Le MH, Devaki P, Ha NB, et al. Prevalence of non-alcoholic fatty liver disease and risk factors for advanced fibrosis and mortality in the United States. PLoS One. 2017;12(3):e0173499.

Thank you for these suggestions. We have added them.

4] The median duration of follow-up differed between groups. Does this translate to the severity of the disease in each group? For example, do patients with cirrhosis died earlier? Please comment.

This is an important point to consider. We agree that the results of the cirrhosis group can be attributed to lead time bias. They were older at the start of follow-up and are, by definition, at a more advanced stage of disease. We have added a section in the Discussion to specifically comment on this limitation (lines 378-382).

We do not feel that this limitation applies to NAFL/NASH groups. They are of similar ages and earlier stage disease is generally relatively asymptomatic.

5] Although lifestyle interventions do remain the first-line therapeutic option for NAFLD/NASH, other drugs may be helpful and especially statins and antidiabetic drugs. A brief comment should be added. Useful refs

Athyros VG, Alexandrides TK, Bilianou H, et al. The use of statins alone, or in combination with pioglitazone and other drugs, for the treatment of non-alcoholic fatty liver disease/non-alcoholic steatohepatitis and related cardiovascular risk. An Expert Panel Statement. Metabolism. 2017;71:17‐32.

Stahl EP, Dhindsa DS, Lee SK, Sandesara PB, Chalasani NP, Sperling LS. Nonalcoholic Fatty Liver Disease and the Heart: JACC State-of-the-Art Review. J Am Coll Cardiol. 2019;73(8):948‐963

Thank you for highlighting this. We have added a comment in the Discussion as well as the suggested references (lines 331-335).

6] Drug therapy may affect clinical outcomes. The inability to evaluate drug effects in this study should be mentioned in the limitations.

Katsiki N, Perakakis N, Mantzoros C. Effects of sodium-glucose co-transporter-2 (SGLT2) inhibitors on non-alcoholic fatty liver disease/non-alcoholic steatohepatitis: Ex quo et quo vadimus?. Metabolism. 2019;98:iii‐ix.

Athyros VG, Polyzos SA, Kountouras J, et al. Non-Alcoholic Fatty Liver Disease Treatment in Patients with Type 2 Diabetes Mellitus; New Kids on the Block. Curr Vasc Pharmacol. 2020;18(2):172‐181.

This is an important consideration. We have added a comment in the Discussion as well as the suggested references (lines 335-338).

Reviewer #3:

Why the inclusion period was only until to 2013 (and not until 2018-2019) and why now do you choose to publish these data?

We appreciate that the end of follow-up was several years ago however the process of data extraction to build our database for took two years to complete after acquiring the data in 2014.

We unfortunately do not have the ability to update our data on these individuals due the logistical process of extracting from the NHS systems. We acknowledge the limitation of this and have now commented on it in the Discussion (lines 359-361).

We had initially aimed to publish this data in 2016 and have updated our methodology substantially since then in light of new evidence and further reports on this subject.

In addition, completion of this work has proved challenging for a variety of reasons, including personal/family circumstances.

We appreciate that this delay detracts from the novelty of the results but trust that it still provides a marginal contribution to the literature of on this topic.

The diagnosis of this pathology have suffered modification between 2000 and 2013. Therefore, the codification is not sufficiently to be sure that the patients was appropriately included. What where the modalities of diagnosis for these pathologies? I think that this is the main cornerstone of this study (the inhomogeneity of diagnosis for these pathologies).

We agree that the principal limitation of this work is that we cannot know how each coded diagnosis was identified. Also in response to Comment 1 from Reviewer #4, we have merged the NAFL and NASH groups into a non-cirrhotic NAFLD group. We have now presented all our results in the main text for: Control, non-cirrhotic NAFLD, and NAFLD-cirrhosis. We have included data as NAFL/NASH groups in the Supplementary Material.

In light of your comments, and recent guidance on retrospective identification of NAFLD diagnoses (Noureddin et al., Gastroenterology 2020 159(2):422-427.e1), we have added more stringent exclusion criteria into our methodology. This has led to the exclusion of some patients but reassuringly the findings of our study remains unchanged. This has increased our confidence that the patients described are those with NAFLD and the effect observed are true.

The data comes from comparatively small, district hospitals in the North of England and therefore it is unlikely that many patients will have had advanced imaging (e.g. magnetic resonance spectroscopy or Fibroscan).

We speculate that the majority of participants will have been diagnosed with NAFLD through ultrasound or incidentally on computed tomography (CT), whereas those with NASH will have been diagnosed using liver biopsy.

In table 1, I do not see the statistical significance (p). It would be important.

Thank you for this suggestion. We have added q-values (false-discovery rate-corrected p-values) into all our tables to help describe all the comparisons.

All-cause mortality has the same percentage in controls and NAFL.

Crude all-cause mortality does not differ between controls and NAFL (or NAFLD) however, after adjusting for age, sex, and ethnicity they have increased mortality. The control group is older than the NAFLD cohort, therefore adjustment for age is important in this analysis. Apologies for not having been clear on this initially.

We have added a comment in the Results to highlight this difference and the importance of the adjusted hazard ratio for interpretation of the mortality data (line 263).

In table 2 why do you not put the sixth column for p? It would be easier to follow these data.

We have also added q-values into table 2.

``The cirrhosis group showed higher prevalence of heart failure, atrial fibrillation, CKD, and ischaemic heart disease, compared to the NAFL group.`` I suggest to analyze separately this subgroup of patients. It might offer very interestingly data about the relationship between thrombosis risks in patients with cirrhosis.

We agree that this is an interesting sub-group however, we are unfortunately limited by the depth of data we have available on each participant. If we had further data (e.g. biochemistry, anthropometry, platelet count, coagulation profile), then we would interrogate the characteristics of participants with cirrhosis who did and did not suffer events. The drawback of our real world routinely-collected data is that it doesn’t allow detail for such an analysis.

We have added a short paragraph in the Discussion to acknowledge this as an important topic worthy of further investigation (lines 363-367).

Reviewer #4:

Major:

Populations definition:

1. As authors reported in study limitation section, the use of ICD code to define the presence or the absence of NAFL/NASH, is highly defective. However, some bias could be reduced considering NAFL and NASH as a unique non cirrhotic NAFLD group. Often, asymptomatic NAFLD patients classified as NAFL patients hide NASH patients with normal liver function tests and non-invasive markers of fibrosis.

We agree that this is a key limitation of attempting to separate NAFL and NASH groups, therefore we have now merged the two into a single (non-cirrhotic) NAFLD group. We have updated all our figures and tables accordingly.

We have moved our data on NAFL and NASH groups into the Supplementary material, in order to provide the raw data for readers to review. 

2. The absence of an ICD10 which refers to NAFLD spectrum does not allow to exclude the presence of nafld among controls (as stated in study limitation section). However, the authors should stress that this bias could underestimated the HR in the analysis and not conversely.

Thank you for this suggestion we have now added a couple of sentences in the discussion to reflect this (lines 351-353).

Statistical methods:

3. Since the authors talk about prevalence of liver decompensation/failure or HCC, I suppose that were recorded ad admission in hospital (follow up start time). In statistical methods authors properly say they used Multivariate logistic regression to define factors associated with this outcome. However, they improperly use the “hazard ratio” to define the statistical outcome. In this case, being a no time-corrected method, the statistical outcome is “odds ratio”, please amend properly

Apologies for this inaccuracy. We have now corrected all the tables and data in the text appropriately.

4. Figure 1. I think that the Kaplan Mayer curves, being the graphic representation of an univariate model, are not the best graphic model to represent survival in this study. At follow up start time patients and controls had different prevalence of cardiovascular (and liver-related?) conditions that may affect survival. Please consider using multiple survival curve (controls vs NAFLD (as suggested in point 1) and control vs patients with cirrhosis) derived from cox-regression analyses.

Thank you for the suggestion. We have exchanged our Kaplan-Meier curve for cumulative hazard survival curves derived from the Cox proportional regression models, which more accurately reflect the differences between groups.

5. Why liver decompensation/failure/HCC was not included in cox multivariate regression? The differences in survival may partially been explained by different prevalence of liver complications. An interesting way to test this hypothesis could be an additional cox regression analysis performed after excluding patients whit liver complications.

We have now performed four models for assessment of mortality, with adjustment for: demographics only, demographics + CVD, demographics + liver-related events, and demographics + CVD + liver-related events. This found that adjusting for liver-related events attenuated the difference in mortality between control and non-cirrhotic NAFLD group (HR 1.1 (0.9-1.3)). Apologies for not having included this before now; thank you for the suggestion.

In addition, we have performed the sub-analysis as the Reviewer suggested (Table S4), which similarly found that after removal of these participants there was no difference in mortality between the non-cirrhotic NAFLD and control groups.

Strength of results:

6. Concern reported in point 5 and the inability to define with certain the cause of hospital admission (follow up start time) raise some doubts that NAFLD patients included in this cohort could represents a cluster of most severe patients in which NAFLD plays a key role.

We appreciate that our results are likely to represent a more severe part of all patients with NAFLD. Comparing the rates of liver-related events in our cohort with previously reported data suggests that they are similar to other biopsy-cohorts. However our control population were also hospitalised individuals therefore had some other form of substantial morbidity, so we believe our comparisons within this study are valid. Therefore, we suggest that our results are accurate for the analyses performed but are not generalisable to other patients with NAFLD (e.g. outpatients, asymptomatic individuals in the general population).

We have added a paragraph in our Discussion (lines 369-376) about this limitation of the generalisability of our results and have also edited our Conclusion.

Minor:

1. Lines 124: another study described the CV burden in patients with NAFLD-cirrhosis, please discuss it (https://doi.org/10.1016/j.atherosclerosis.2017.03.038)

Thank you for highlighting this important study to us. We have now commented on it in the Introduction and Discussion (lines 116-118).

2. Table 1: Adjusted and unadjusted HRs for overall mortality and ORs (?) for Liver failure/decompensation and HCC should be reported in a separate table.

We have now re-arranged our tables into three: demographics, crude disease/event rates, and adjusted HR/ORs.

3. The sentence “increased mortality for patients with NAFLD, irrespective of fibrosis” (lines 277-278) is not supported by results. In this study, the only differentiation in the severity of fibrosis could be made comparing cirrhotic with non-cirrhotic patients and, among them there is a wide difference in survival (HR 3.8 vs 2.6 vs. 65.4!)

Apologies for this inaccuracy. We had intended to say that our non-cirrhotic NAFLD group also had increased mortality compared to controls. We have edited this section (line 288) to make it more accurate.

4. Lines 298-304: a recent study proved higher incidence of CV disease in patients with NAFLD compared to metabolic controls. In addition, in the same study, non-invasive markers of fibrosis identify patients at higher risk for CV events among those with NAFLD. The same study could be discussed in introduction section (lines 101-107) to affirm that fibrosis, in NAFLD patients, is the major predictors of CVE also. doi: 10.1016/j.cgh.2019.12.026

Thank you for highlighting this important study, which has come out since the original version of our paper was submitted. We have now commented on it in both the Introduction and Discussion (lines 312-317).

5. Please check through the paper the correct use of NAFLD/NAFL terms. Eg: in figure 1 “NAFLD” was incorrectly used instead of NAFL

Apologies for these typographical errors. We have carefully checked the paper and made corrections.

---

## [Decision Letter · Decision Letter 1]

14 Oct 2020

Hospital admission with non-alcoholic fatty liver disease is associated with increased all-cause mortality independent of cardiovascular risk factors

PONE-D-19-32801R1

Dear Dr. Mann,

We’re pleased to inform you that your manuscript has been judged scientifically suitable for publication and will be formally accepted for publication once it meets all outstanding technical requirements.

Kind regards,

Michael W Greene, Ph.D.

Academic Editor

PLOS ONE

Reviewers' comments:

Reviewer's Responses to Questions

**Comments to the Author**

1. If the authors have adequately addressed your comments raised in a previous round of review and you feel that this manuscript is now acceptable for publication, you may indicate that here to bypass the “Comments to the Author” section, enter your conflict of interest statement in the “Confidential to Editor” section, and submit your "Accept" recommendation.

Reviewer #1: All comments have been addressed

Reviewer #2: All comments have been addressed

Reviewer #3: All comments have been addressed

2. Is the manuscript technically sound, and do the data support the conclusions?

Reviewer #1: Yes

Reviewer #2: Yes

Reviewer #3: Yes

3. Has the statistical analysis been performed appropriately and rigorously? 

Reviewer #1: Yes

Reviewer #2: Yes

Reviewer #3: Yes

4. Have the authors made all data underlying the findings in their manuscript fully available?

Reviewer #1: Yes

Reviewer #2: Yes

Reviewer #3: Yes

5. Is the manuscript presented in an intelligible fashion and written in standard English?

Reviewer #1: Yes

Reviewer #2: Yes

Reviewer #3: Yes

6. Review Comments to the Author

Reviewer #1: This is a well written paper with results that have clinical implications for the treatment of patients with NAFLD-cirrhosis.

The research paper PONE-D-19-32801 entitled “Hospital admission with non-alcoholic fatty liver disease is associated with increased all-cause mortality independent of cardiovascular risk factors” has relevance to the scope and the audience of PONE.

This paper aimed to assess the mortality in NAFLD when adjusting for CVD risk factors in a ‘real world’ cohort of inpatients.

The authors suggest that there is a high burden of cardiovascular disease in NAFLD-cirrhosis patients. From a large “real-life” non-specialist registry of hospitalized patients, NAFLD patients have increased overall mortality and rate of liver-related complications compared to controls after adjusting for cardiovascular disease.

Major comments for the authors

1. This analysis was performed on an interesting issue: The effect of NAFLD on overall mortality in a real world setting regardless of other CVD risk factors.

2. It is a well written paper. The text, the figure and the tables are of appropriate extent and content as well as informative.

3. The references are up to date.

4. The results of the paper have practical implications in patients with NAFLD, especially those with cirrhosis.

5. Any issues with the paper in the original submission were addressed in a proper way in the revision

Reviewer #2: (No Response)

Reviewer #3: (No Response)

7. PLOS authors have the option to publish the peer review history of their article (what does this mean?). If published, this will include your full peer review and any attached files.

Reviewer #1: No

Reviewer #2: No

Reviewer #3: No

---

## [Editor Report · Acceptance letter]

16 Oct 2020

PONE-D-19-32801R1 

Hospital admission with non-alcoholic fatty liver disease is associated with increased all-cause mortality independent of cardiovascular risk factors 

Dear Dr. Mann:

I'm pleased to inform you that your manuscript has been deemed suitable for publication in PLOS ONE. Congratulations! Your manuscript is now with our production department. 

Kind regards, 

on behalf of

Dr. Michael W Greene 

Academic Editor

PLOS ONE